

# Evaluation of Anthropogenic Secondary Organic Aerosol Tracers

Ibrahim M. Al-Naiema, Elizabeth A. Stone

Department of Chemistry, University of Iowa, Iowa City, IA, 52242, USA

*Correspondence to*: Elizabeth A. Stone (betsy-stone@uiowa.edu)

**Abstract.** Products of secondary organic aerosol (SOA) from aromatic volatile organic compounds (VOC)—2,3-dihydroxy-4-oxopentanoic acid, dicarboxylic acids, nitromonoaromatics, and furandiones—were evaluated for their potential to serve as anthropogenic SOA tracers with respect to their 1) ambient concentrations and detectability in $PM_{2.5}$ in Iowa City, IA, USA, 2) gas-particle partitioning behaviour, and 3) source specificity by way of correlations with primary and secondary source tracers and literature review. A widely used tracer for toluene-derived SOA, 2,3-dihydroxy-4-oxopentanoic acid was only
detected in the particle phase ($F_p = 1$) at low, but consistently measureable ambient concentrations (averaging 0.3 ng m$^{-3}$). Four aromatic dicarboxylic acids were detected at relatively higher concentrations (9.1 - 34.5 ng m$^{-3}$), of which phthalic acid was the most abundant. Phthalic acid had a low particle-phase fraction ($F_p = 0.26$) likely due to quantitation interferences from phthalic anhydride, while 4-methylphthalic acid was predominantly in the particle phase ($F_p = 0.82$). Phthalic acid and 4-methyl phthalic acid were both highly correlated with 2,3-dihydroxy-4-oxopentanoic acid ($r_s$=0.73, p=0.003; $r_s$=0.80,
p<0.001, respectively), suggesting that they were derived from aromatic VOC. Isophthalic and terephthalic acids, however, were detected only in the particle phase ($F_p = 1$) and correlations suggested association with primary emission sources. Nitromonoaromatics were dominated by particle-phase concentrations of 4-nitrocatechol (1.6 ng m$^{-3}$) and 4-methyl-5-nitrocatechol (1.6 ng m$^{-3}$) that were associated with biomass burning. Meanwhile, 4-hydroxy-3-nitrobenzyl alcohol was detected in a lower concentration (0.06 ng m$^{-3}$) in the particle phase only ($F_p = 1$), and is known as a product of toluene
photooxidation. Furandiones in the atmosphere have only been attributed to the photooxidation of aromatic hydrocarbons, however the substantial partitioning toward the gas phase ($F_p \leq 0.16$) and their water sensitivity limit their application as tracers. The outcome of this study is the demonstration that 2,3-dihydroxy-4-oxopentanoic acid, phthalic acid, 4-methylphthalic acid, and 4-hydroxy-3-nitrobenzyl alcohol are good candidates for tracing SOA from aromatic VOC.




## 1 Introduction

Secondary organic aerosol (SOA) accounts for a large but undefined fraction of organic matter in $PM_{2.5}$, forming through the photooxidation of biogenic and anthropogenic volatile organic compounds (VOCs) in the gas phase yielding low-vapour pressure products that partition into the particle phase (Kroll and Seinfeld, 2008; Hallquist et al., 2009). The

global fluxes of anthropogenic SOA is poorly constrained and highly uncertain, with a wide range of estimates from 2-25 Tg $yr^{-1}$ (Volkamer et al., 2006; Henze et al., 2008). Measurements suggest that anthropogenic precursors form more SOA than predicted by models (Heald et al., 2005; Volkamer et al., 2006; Matsui et al., 2009), likely due to incomplete model representation of SOA formation pathways (Henze et al., 2008), partitioning (Donahue et al., 2006), ambient conditions (Ng et al., 2007), and precursors (Robinson et al., 2007; Fu et al., 2008).

A tracer-based approach has been useful in identifying aerosol sources and source apportionment (Schauer et al., 1996). SOA can be linked to its precursor VOC following the SOA tracer approach introduced by Kleindienst et al. (2007) in which ambient concentration of tracers (or the sum of thereof) are converted to secondary organic carbon (OC) or SOA mass yields using tracer-to-OC or tracer-to-SOA ratios, respectively, that were determined in chamber studies. For biogenic SOA, relatively-well defined and established tracers are employed, such as methyltetrols for isoprene and β-caryophyllinic

acid for β-caryophyllene (Kleindienst et al., 2007). In contrast, the tracer-based approach for aromatic SOA relies on a single molecule (2,3-dihydroxy-4-oxopentanoic acid [DHOPA]) that is derived from toluene (Kleindienst et al., 2007). Advancing the tracer-based approach to anthropogenic SOA apportionment should involve expanding the number of available tracers, particularly those to form from aromatic VOC other than toluene.

Chamber experiments have been conducted to identify SOA products formed during the photooxidation of aromatic

precursors associated with anthropogenic sources, such as benzene, toluene, ethylbenzene, xylenes (BTEX) and low molecular weight polycyclic aromatic hydrocarbons (PAHs). Furandiones have been identified as a major product of aromatic VOC photooxidation in the presence of $NO_x$ (Bandow et al., 1985; Forstner et al., 1997; Hamilton et al., 2003; Koehler et al., 2004). Nitromonoaromatics (e.g., nitrophenols, methyl-nitrophenols, nitrocatechols, and nitrosalicylic acids) are likewise products of aromatic VOC photooxidation in the presence of $NO_x$ (Forstner et al., 1997; Jang and Kamens,

2001; Hamilton et al., 2005; Sato et al., 2012; Irei et al., 2015), but some species have also been detected in the primary emission from vehicles (Tremp et al., 1993) and biomass burning (Iinuma et al., 2010). While nitromonoaromatics have been quantified in ambient aerosol (Dron et al., 2008; Kitanovski et al., 2012; Kahnt et al., 2013), the extent of their formation from primary and/or secondary sources has yet to be determined. Phthalic acid is a product of naphthalene photooxidation (Kautzman et al., 2010), and proposed as a tracer for naphthalene and methylnaphthalenes in $PM_{2.5}$ (Kleindienst et al., 2012).

However, phthalic acid has also been observed in the emission of motor exhaust (Kawamura and Kaplan, 1987), and thus both primary and secondary sources can contribute to its ambient concentration. These three classes of compounds can be potentially used a tracers for SOA; however, further ambient studies are needed to evaluate their detectability, ambient concentrations, and origins.





There are many desired characteristics for a molecule to be used as a source tracer. First, it should be unique to the source of origin. For example DHOPA was previously identified as a unique product of toluene photooxidation in the presence of $NO_x$ (Kleindienst et al., 2004) and methyltetrols are unique to isoprene (Claeys et al., 2004a). Second, the tracer should be formed in reasonably high yields so it has sufficiently high concentrations in the atmosphere to allow for reliable

quantification. Third, the tracer needs to be reasonably stable in the atmosphere, so that it is conserved between formation and collection at a receptor location. Fourth, an efficient SOA tracer should have a low vapour pressure so that it primarily partitioned to the particle phase, which minimizes possible underestimation from loss to the gas phase. Thus, an effective SOA tracer will exhibit source specificity, consistent detectability, atmospheric stability and partitioning to the aerosol phase.

In this work, we examine and evaluate the efficacy of nitromonoaromatics, furandiones, and aromatic dicarboxylic acid isomers as potential SOA tracers in terms of their ambient concentration, gas-particle partitioning, and source specificity through correlations with established tracers, including levoglucosan for biomass burning (Simoneit et al., 1999), hopanes for vehicular emissions (Schauer et al., 1999), and DHOPA for anthropogenic SOA (Kleindienst et al., 2004). Sample preparation procedures were optimized for the simultaneous extraction of primary and potential secondary source tracers,

which were then quantified by GCMS. These methods were applied to measure the ambient concentrations and gas-particle distributions for analytes in fine particulate matter ($PM_{2.5}$) collected in Iowa City, IA in the fall of 2015. Developing and evaluating these tracers provides additional tools for better understanding the contribution of aromatic VOC to ambient aerosol and will help to expand the current knowledge about the composition and sources of ambient aerosol, particularly in urban and peri-urban environments.

**2 Experimental methods**

**2.1 Field Sampling**

Gas and particle ($PM_{2.5}$) samples were simultaneously collected daily for the period 4-15 November 2015 in Iowa City, IA, USA (41.6572° N, 91.5035° W). The sampler was installed on a wooden platform, and the inlet was positioned 3.5 m above ground level. The sampling site was surrounded by an agricultural field and a university parking lot. Sample

collection was preformed using a medium-volume URG air sampler (3000B, URG Corp.) with a cyclone (URG) operating at a flow rate of 90 L min$^{-1}$. Air flow rate was monitored before and after sampling using a rotameter (Gilmont Inst.). $PM_{2.5}$ samples were collected on 90-mm quartz fiber filters (Pallflex® Tissuquartz™, Pall life science) that were pre-cleaned by baking for 18 hours at 550 °C. Gas samples were collected on 52 mm polyurethane foam (PUF) plug placed after the filter holder (URG-2000-30-52PC). PUF plugs were pre-cleaned using acetone (HPLC grade, Sigma-Aldrich), hexanes, and

acetonitrile (Optima-Fisher Scientific-Fisher Chemical) by a repeated compression extraction apparatus adapted from Rogge and Ondov (2011). This apparatus is composed of a thick-walled borosilicate glass cylinder equipped with polytetrafluoroethylene (PTFE) valve and PTFE plunger that was used to compress the solvent out of the PUF.



Samples were collected for 23 hours, and filter changing was performed at 8:00 am (local time). After sampling, filters were transferred to Petri dishes, lined with pre-baked aluminum foil, and sealed with Teflon tape. PUF samples were transferred to a pre-baked wide-mouth glass jar, capped with a Teflon-lined cap, and sealed with Teflon tape. Sampled filters and PUF were transported to the laboratory and stored frozen at -20 $^\circ$C until analysis. One field blank was collected for every five samples following the same described procedure, except no air was pulled through the system.

## 2.2 Extraction

All glassware used in this experiment was first baked (500 °C for 5 hours) to remove organic contaminants and then silanized (using 5% solution of dichlorodimethylsilane in toluene) to minimize the sorption of analytes to the glass surface (Kitanovski et al., 2012). Filters and PUF were spiked with isotopically-labelled internal standards, representing the different classes of organic compounds reported in this study. Adding internal standards prior to extraction corrects for loss of analyte during the extraction process, provided the internal standard adequately represents the chemical and physical properties of the analyte. Specifically, internal standards and their corresponding analytes were 3-nitrosalicylic acid-D3 and 5-nitrosalicylic acid, 4-nitrophenol-D4 and other nitromonoaromatics, maleic anhydride-$D_2$ for 2,5-furandione, and succinic anhydride-2,2,3,3-$D_4$ for the three other furandiones, levoclucosan-C13 for levoglucosan, ketopinic acid for DHOPA, phthalic acid-D4 for aromatic dicarboxylic acids, and acenaphthene-D10, pyrene-D10, benzo[a]anthracene-D12, coronene-D12 for PAHs.

Filters were extracted sequentially with three 10 mL portions of acetonitrile by ultra-sonication (60 sonics per minute, for 15 min). The combined three extracts were reduced to 2 mL by rotary-evaporation at 30 °C, 120 rpm, and 200 mbar. The reduced extracts were filtered with a 0.25 μm PTFE syringe filters and stored frozen at − 20 °C until analysis. Immediately prior to analysis, the extracts were evaporated to 100 μL under a gentle stream of ultra-pure nitrogen at 30 °C. PUF samples were extracted by three-cycles of repeated compression using acetonitrile; extracts were then combined, evaporated, filtered, and reduced to the final volume using the same conditions as filter extracts.

## 2.3 Instrumental analysis

OC and elemental carbon (EC) were measured by thermal-optical analysis (Sunset Laboratory Inc.) on a 1 cm$^2$ filter portion, following Schauer et al., (2003).

Organic species were analysed using an Agilent 7890A GC, coupled with 5975C MS (Agilent Technologies). 2,3-Dihydroxy-4-oxopentanoic acid (Toronto Research Chemicals), phthalic acid isomers, levoglucosan, and biogenic SOA tracers were trimethylsilylated with *N,O*-bis(trimethylsilyl)trifluoroacetamide (Fluka Analytical 99%) prior to analysis. The silylation reaction was performed by evaporating 10 μL of the extract under a gentle stream of nitrogen to dryness, adding 20 μL of the silylation agent and 10 μL of pyridine (Burdick & Jackson, Anhydrous), and heating to 70 °C for 3 hours. A 2 μL aliquot of the silanized extract was introduced to the GCMS equipped with DB-5 column, electron impact (EI) ionization source (70 eV), with a GC inlet temperature of 300 °C. Nitromonoaromatics were also silylated using the same agent, but





under different conditions in which a 20 µL of the extract was evaporated to dryness under a gentle stream of nitrogen, 10 µL of the silylation agent was added, and then the mixture was then capped and heated for 90 min at 100 °C. The GC injection volume was 1 µL and while the inlet conditions, column type, and MS parameters matched those previously described. Furandiones were analysed using the method developed in our previous work (Al-Naiema et al., in review), and

PAHs were analysed using DB-5 column as described elsewhere (Al-Naiema et al., 2015).

Responses of the analytes were normalized to the corresponding isotopically-labelled internal standards, and quantified using linear calibration curve with a squared correlation coefficient ($R^2$) ≥ 0.995. Analytical uncertainties were propagated from the standard deviation of the field blank value and 10% of the measured concentration. For analytes not detected in field blank, instrument detection limits were used in error propagation. All measurements were field blank

subtracted. Due to low recoveries of furandiones from PUF, gas-phase concentrations of furandiones were corrected for the recoveries of the authentic standards.

### 2.4 Particle-phase fraction calculation and model

The fraction of a species in the particle phase ($F_p$) was calculated from the ratio of concentration in the particle-phase to the total concentration (sum of gas and particle), following Eq. (1).

$$F_p = \frac{[particle]}{[gas]+[particle]} \tag{1}$$

$F_p$ was modelled using the gas-particle partitioning coefficient (Eq. 2) from absorptive partitioning theory developed by Pankow (1994), and following Yatavelli et al., (2014):

$$F_p = \left(1 + \frac{1}{k_{om} \times C_{OA}}\right)^{-1} \tag{2}$$

Where $C_{OA}$ is concentration of the organic aerosol (µg m$^{-3}$), and $k_{om}$ is the partitioning coefficient (m$^3$ µg$^{-1}$) described as:

$$k_{om} = \frac{RT}{10^6 \, P_L^0 \, \mathcal{E} \, MW} \tag{3}$$

in which R is ideal gas constant (8.2 × 10$^{-5}$ m$^3$ atm mol$^{-1}$ K$^{-1}$), T is temperature (averaging 9 °C during this study), 10$^6$ is unit conversion factor (µg g$^{-1}$), $P_L^0$ is the sub-cooled vapour pressure (atm), obtained from the Estimation Program Interface suit$^{TM}$ version 4.11 from the Environmental Protection Agency (EPA), MW is the molar mass (g mol$^{-1}$), and $\mathcal{E}$ is the activity coefficient (assumed to be 1).

### 2.5 Statistical analysis

Inter-species correlation were evaluated using Minitab software (version 16). The Anderson-Darling test for normality indicated that neither ambient concentrations nor log-transformed concentrations were normally distributed. Hence, Spearman's rho ($r_s$) was used to assess correlations. Correlations were interpreted as follows: very high (0.9-1.0, high (0.7-0.9), moderate (0.5-0.7), low (0.3-0.5), and negligible (0.0-0.3) (Mukaka, 2012). The statistical significance of

correlations was evaluated at the 95% confidence interval (p ≤ 0.05).





# 3 Results and discussion

## 3.1 Validation of gas-particle partitioning

PAHs with two to eight rings span a range of high to low volatility, respectively. The accuracy of the measured gas-particle distributions were evaluated with PAHs that have been extensively discussed in the literature. The average fractions of PAH in the particle phase ($F_p$) measured in Iowa City, IA, USA (Fig. 1) were ~5% for 10-14 carbon atoms (2-3 rings: naphthalene, acenaphthene, and anthracene), 14% for 16 carbon atoms (4 rings: pyrene), 59% for 18 carbons (4 rings: benzo(ghi)fluoranthene), and > 98% for 20 carbon atoms (5 rings: picene). The predicted $F_p$ values estimated using Pankow absorption model (1994) following Eq. (1) and (2), and using parameters in Table S1 follow the same trend (Fig. 1, dashed line), with a systematic underestimation for the predicted $F_p$ by ≤ 12% for most PAH and 20% for 18 carbons. Such an underestimation has been widely documented in comparison of theory to ambient partitioning studies, and is attributed to the omission of PAH sorption on elemental carbon in the model (Dachs and Eisenreich, 2000; He and Balasubramanian, 2009; Wang et al., 2011). Ambient studies of gas particle partitioning are influenced by many factors such as ambient temperatures (Terzi and Samara, 2004), relative humidity (Pankow et al., 1993), and sampling technique (reviewed by Kim and Kim, 2015), confounding direct comparisons between this and other studies. Overall, the general trends observed herein are consistent with prior studies that report PAHs with 2-3 aromatic rings ($C_{10}$-$C_{14}$) mainly in the gas phase ($F_p ≥ 0.93$), those with 5 or more aromatic rings ($≥ C_{20}$) mainly in the particle phase ($F_p ≥ 0.9$), and those with 4 aromatic rings ($C_{16}$ and $C_{18}$) partition between the two phases depending of their chemical structure and atmospheric conditions (Yamasaki et al., 1982; Williams et al., 2010; Ma et al., 2011; Kim and Kim, 2015).

Gas phase sampling using QFF without a denuder upfront is a subject to artefacts caused by vapour adsorption on the filter, resulting in underestimation for the concentration of the species measured in the gas phase, particularly for low MW PAHs (Delgado-Saborit et al., 2014). However, comparing the partitioning trend in this study to those sampled with a denuder during the same season (Possanzini et al., 2004) shows less than 5% discrepancies for the low (C ≤ 12) and high (C ≥ 18) molecular weight PAHs, while for $C_{14}$ and $C_{16}$, our $F_p$ measurements were lower by 14% and 6%, respectively. These results show a slight underestimation of $F_P$ rather than an overestimation that would be expected if vapour adsorption on the QFF significantly impacted gas-particle partitioning results. We estimate that the uncertainties associated with our gas-particle partitioning measurements are ≤ 5% for species predominantly in the particle phase ($F_P > 0.9$) or gas phase ($F_P < 0.1$) and are in the range of 14% for semi-volatile species ($0.1 < F_P < 0.9$).

## 3.2 The toluene tracer (2,3-dihydroxy-4-oxopentanoic acid)

### 3.2.1 GC-MS identification

2,3-Dihydroxy-4-oxopentanoic acid (DHOPA, also known as T-3) has been identified as a product of toluene photooxidation by Kleindienst et al., (2004) and their chemical ionization mass spectrum has been used to identify this tracer in other studies. To support identification by the more common electron ionization (EI), the corresponding mass spectrum of



its trimethylsilylated (TMS) derivative is given in Fig. 2. The most abundant ions are m/z 73 and 147 corresponding to $Si^+(CH_3)_3$ and $(CH_3)_2Si=O^+Si(CH_3)_3$ fragments, respectively; however, these are common to the BSTFA-TMCS silylation reagent. Ions at m/z 277, 349, 321, 364 are unique to DHOPA and are recommended for quantification. Here, m/z 277 ion was used for quantification due to the high relative abundance and low background, and the other ions were used as qualitatively. This mass spectrum obtained from a pure standard builds upon the previous EI-mass spectrum for the DHOPA in an aerosol sample by Hu et al. (2008) that included some spectral interferences from adipic acid that co-eluted.

### 3.2.2 Ambient concentration and gas-particle partitioning

The average mass concentration of DHOPA ranged from 0.14 to 0.50 ng m$^{-3}$ and averaged 0.29 ± 0.12 ng m$^{-3}$ (Fig. 3a). DHOPA was detected only in the particle phase (Table 2), although the 22.1 ± 13.5 % extraction recoveries of this species from PUF limited the sensitivity of gas phase measurements. Nonetheless, it is reasonable to conclude that this species does not appreciably partition to the gas phase.

The average concentrations of DHOPA in Iowa City were within the range of those observed in Bondville, IL in fall (Lewandowski et al., 2008), but were lower by a factor of 45 (on average) than what was detected in the Pearl River Delta, China for the same season (Ding et al., 2012). Although an authentic standard for DHOPA was not previously available, prior measurements based upon a surrogate standard response and are subject to bias.

The contribution of toluene SOA to OC was estimated based on the SOA-tracer method introduced by Kleindienst et al., (2007), where DHOPA accounted for 0.0079 ± 0.0026 of secondary OC mass from toluene. Following this estimation method, toluene SOC was estimated to contribute 36.5 ± 15.0 ngC m$^{-3}$. The contribution of the estimated SOC to the total OC in this study ranged 0.3-7% and averaged 2.2 ± 1.6%. In other studies, the concentration of toluene SOC was variable and influenced by seasonal variations and local emission sources (Kleindienst et al., 2007; Peng et al., 2013). Our estimated SOC levels were less than half of those observed in the rural Midwestern United States previously (0.09 µg m$^{-3}$) during the same season; however the contribution of the estimated toluene SOC to the total OC was 6% (Lewandowski et al., 2008), which is within the upper end of the range observed in this study. Although toluene SOC concentrations were much higher in the Pearl River Delta (1.65 µg m$^{-3}$) (Ding et al., 2012), the relative contribution to OC (7%) was comparable. Because toluene is only one of many aromatic VOC precursors to SOA, additional tracers are needed to better evaluate the impact of aromatic VOC on SOA.

### 3.3 Benzene dicarboxylic acids

Three isomers of benzene dicarboxylic acid and one methyl derivative were detected in all PM samples. The total (gas plus particle) concentration of phthalic acid (PhA) , the most abundant isomer, ranged from 4.9 - 21.5 ng m$^{-3}$, and averaged 13.0 ± 4.3 ng m$^{-3}$, while isophthalic acid (i-PhA), terephthalic acid (t-PhA) and 4-methylphthalic acid (4M-PhA) had increasingly lower concentrations in the range of 0.2 to 6.6 ng m$^{-3}$ (Fig. 3b). Similar relative abundancies for these species were observed in other studies, with PhA consistently being the predominant isomer (Fraser et al., 2003; Mirivel et





al., 2011; Mkoma and Kawamura, 2013). The relatively high ambient mass concentrations of these dicarboxylic acids isomers at levels that allow for consistent detection makes them promising candidates for tracing aromatic SOA.

The majority of PhA was estimated to be in the gas phase ($F_P = 0.26$), in contrast to i-PhA and t-PhA ($F_P = 1$) and 4-M-PhA ($F_P = 0.82$, Table 2). Vapour pressure values and partitioning theory (Table S1) cannot explain the observed lower fraction of PhA in the particle phase compared to i-PhA and t-PhA ($F_P = 1$) . Instead, the gas phase measurement of PhA is expected to be positively biased due to interference by phthalic anhydride, which yields identical products to PhA when derivatized, hydrolysed or exposed to high temperatures (like those encountered in GC analysis). For example, under the conditions employed in this study, phthalic anhydride and PhA have identical GC retention time and silylated MS spectra (Fig. S1). Phthalic anhydride is a gas phase product of naphthalene photooxidation (Chan et al., 2009; Kautzman et al., 2010) and has much higher vapour pressure ($7.5 \times 10^{-6}$ atm) than that of PhA ($8.9 \times 10^{-8}$ atm) (EPA, 2012).  As such, phthalic anhydride will partition to a greater extent to the gas phase, which is supported by the absorption model estimations ($F_P = 4.9 \times 10^{-5}$), shown in Table S1. Thus, PhA concentrations reported here and in prior studies that involve the use of GC inlet temperatures $\geq 150$ °C, derivatization, or hydrolysis (which is common in liquid chromatography) reflect the sum of PhA and phthalic anhydride. Because phthalic anhydride is primarily in the gas phase, this causes gas phase PhA concentrations to be overestimated and $F_P$ estimates for PhA to be erroneously low. An accurate determination of $F_P$ for PhA requires collection and analysis of acid and anhydrides separately by in situ derivatization on veratrylamine-coated glass fiber filters (OSHA, 1991).

Although PhA and 4-M-PhA can be emitted directly from primary sources such as motor vehicle engines (Kawamura and Kaplan, 1987), there is a lack of evidence for significant primary source contributions to these species in ambient air. In contrast, naphthalene, a precursor for secondary formation of PhA (Kautzman et al., 2010; Kleindienst et al., 2012), was found to be the most abundant PAH from many combustion sources (Oanh et al., 1999; Al-Naiema et al., 2015). As shown in Table S3, the concentrations of PhA and 4M-PhA in the particle phase are highly and significantly correlated with DHOPA (rs=0.73, p=0.003) and (rs=0.79, p=0.001) respectively, but they do not correlate with hopane (rs=0.19, p=0.529), a fossil fuel combustion biomarker. These correlation data indicate that the probable origin of these two acids is secondary reactions, rather than primary emissions. Although isophthalic acid has a strong correlation with DHOPA, it also correlates highly and significantly with biomass burning products (e.g., levoglucosan, 4-nitrocatechol, and 4-methyl-5-nitrocatechol) and moderately with hopane. The possibility of multiple sources of i-PhA limits its application as a tracer for anthropogenic SOA. Terephthalic acid correlates strongly with biomass burning tracers and with hopane, and there is no evidence supporting secondary formation; hence, t-PhA is not a valid SOA tracer candidate.  Together, the relative high concentration detected in the particle phase relative to other tracers, and the high correlations with DHOPA suggest that PhA and 4M-PhA are useful SOA tracers for naphthalene photooxidation.



### 3.4 Nitromonoaromatic compounds

#### 3.4.1 Analytical method performance

While many techniques for quantifying nitrophenols in various sample matrices have been developed and were reviewed by Harrison et al, (2005), our goal was to quantify these compounds in parallel to other primary and secondary source tracers using GCMS and single filter extraction protocol. Using GC with a DB-5 column, a baseline separation was achieved for ten nitromonoaromatic analytes as trimethylsilylated esters (Fig. 4), with highly symmetrical and narrow peak shapes. Mass spectrometry was used for identification by comparison of retention times and mass spectra, and quantification was done based on base peak area (Table S2). Save for nitroguiacols, nitromonoaromatics mass spectra included mass fragments of $[M-57]^+$ (loss of $NO_2$ and $CH_3$) and $[M-129]^+$ (loss of $(Si(CH_3)_3)$) for the singly and doubly derivatized analytes, respectively. Nitroguaiacol isomers had a fragment at $[M-42]^+$ (loss of $NO_2$).

The performance of the GCMS method was evaluated with respect to linearity, detection and quantification limits of target analytes, and extraction efficiency (Table 1). The normalized response for the nitromonoaromatics was linear ($R^2 \geq$ 0.996) from 10-5000 µg L$^{-1}$ with a constant internal standard concentration of 10,000 µg L$^{-1}$. This wide range of linearity indicates the suitability of this method to determine nitromonoaromatics in different applications. The limit of detection ranges from 2.7 to 14.9 µg L$^{-1}$ for the ten nitromonoaromatic compounds. These method detection limits are higher than those obtained from liquid chromatography coupled with tandem mass spectrometry (0.1- 0.25 µg L$^{-1}$) (Kitanovski et al., 2012), but are sufficient to detect the investigated species in ambient air. Filter extraction recoveries averaged 99.4 ± 3.8%, demonstrating high accuracy and precision of the filter extraction with acetonitrile and reduction in volume under reduced pressure with rotary evaporation.  For PUF extraction, very high recovery (> 97%) was achieved for most compounds (Table 1), however two nitrocatechols, 4NC and 4M-5NC, had significantly lower (< 50%) and much more variable (RSD 34-60%) recoveries. Similarly low recoveries of 4NC and 4M-5NC have been reported previously (Hawthorne et al., 1989), which is attributed to the strong interactions of phenols with the polymeric chains of the PUF. Consequently, gas phase measurements of 4NC and 4M-5NC are biased low and subject to high uncertainty, such that their levels and gas-particle partitioning are not reported. Otherwise, the extraction and analysis method provides high accuracy and reliable precision for nitromonoaromatics from filters and PUF.

#### 3.4.2 Ambient concentration, gas particle partitioning, and potential sources

Total concentrations of eight nitromonoaromatics ranged from 0.7 - 17 ng m$^{-3}$ in the particle phase, and from 0.6 - 40 ng m$^{-3}$ in the gas phase (Fig. 5 and S3). Average concentrations and $F_P$ are summarized in Table 2, with daily $F_P$ shown in Fig. S2.

A number of nitromonoaromatics were likely derived from biomass burning, as evidenced by correlations with a biomass burning marker (levoglucosan) in this and prior studies. Nitrocatechols were the most abundant particle-phase species within this compound class, with average concentrations (± standard deviation) of 1.6 ± 2.9 ng m$^{-3}$ and 1.6 ± 1.8 ng



$m^{-3}$ for 4NC and 4M-5NC, respectively. These two species have been previously associated with biomass burning in $PM_{10}$, via their correlations the biomass burning marker levoglucosan (Iinuma et al., 2010; Kahnt et al., 2013). The strong correlation of these two species with levoglucosan extends to $PM_{2.5}$ in Iowa City (Fig. 5) with very high correlations with levoglucosan for 4NC ($r_s$=0.90, p<0.001) and 4M-5NC ($r_s$=0.85, p<0.001). Although nitrocatechol can be formed from the

5 toluene photooxidation (Lin et al., 2015), 4NC correlates weakly with DHOPA ($r_s \leq 0.2$) lacking statistical significance (Table S3), suggesting that toluene photooxidation is negligible in relation to biomass burning. Similarly, 5NSA ($F_p = 0.73$) was highly correlated with levoglucosan ($r_s$=0.76, p=0.002), but moderately with DHOPA ($r_s$=0.49, p=0.078), also suggesting its primary origin to be biomass burning (Kitanovski et al., 2012; Zhang et al., 2013) rather than photooxidation (Jang and Kamens, 2001) in agreement with prior studies. Consequently, these three species are characteristic of biomass

burning, rather than anthropogenic SOA.

Nitroguaiacol was detected in low concentrations relative to other nitromonoaromatics. The concentrations of 4NG in $PM_{2.5}$ ranged from below the detection limit (BDL) to 0.11 ng $m^{-3}$, with a frequency of detection of 86% (Table 2). Similarly low concentrations were also reported elsewhere (Kitanovski et al., 2012). In the gas phase, 4NG was not detected on most of the days, except for 14-16 November when gas concentrations reached 0.5-2.1 ng $m^{-3}$ (Fig. S2). On 14

November, outdoor festivities, barbecues, and slow moving traffic occurred near the sampling site. The possibility of multiple sources and the low ambient concentrations suggest that this is not a suitable tracer for anthropogenic SOA.

4H-3NB was detected only in the particle phase ($F_p = 1$) with a frequency of 71% and relatively low concentrations ranging from BDL to 0.2 ng $m^{-3}$. 4H-3NB was identified as a low abundance product of toluene photooxidation with hydroxyl radicals (Hamilton et al., 2005; Fang et al., 2011). Other than toluene photooxidation (Hamilton et al., 2005), there

are no other known emission sources for 4H-3NB. The specificity of 4H-3NB is supported by the lack of correlation with other biogenic or anthropogenic tracers (Table S3). Because of its detected only in the particle phase and is likely specific to toluene photooxidation, it has potential to be a unique nitromonoaromatic tracer for anthropogenic VOCs photooxidation. However, due to the small number of samples and the frequency of detection for this tracer, further investigation is recommended to evaluate its detectability in other environments and source specificity.

In addition, 4-nitrophenol (4NP) was consistently detected, with summed gas and particle concentration of 4NP ranging from 0.3 to 7.3 ng $m^{-3}$ and averaging 1.8 ± 2.1 ng $m^{-3}$. Likewise, two methyl-nitrophenol isomers (4M-2NP and 2M-4NP) were levels averaged 0.3 ± 1.6 ng $m^{-3}$ and 5.3 ± 8.5 ng $m^{-3}$, respectively (Table 2 and Fig. S3). The higher concentration of 4M-2NP with the higher standard deviation is largely driven by the aforementioned local source influences on 14 November (32.5 ng $m^{-3}$), shown in Fig. S2. These three nitromonoaromatics showed a substantial partitioning in the

gas phase, with $F_P \leq 0.33$ (Table 2). The very high correlation of 4NP with 2M-4NP ($r_s$=0.90, p<0.001) and with 4M-2NP ($r_s$=0.81, p<0.001) indicate a similar source of origin. These three compounds have previously been shown to be products from the photooxidation of the monoaromatic compounds in the presence of $NO_x$ (Forstner et al., 1997; Harrison et al., 2005; Sato et al., 2007) as well as components of vehicle emissions (Tremp et al., 1993). However, no significant correlations were





observed between these tracers with hopane or DHOPA. Because of their lack of source specificity and the significant partitioning in the gas phase, these three nitrophenols are not recommended for use as tracers of anthropogenic SOA.

**3.5 Furandiones**

Ambient gas and particle concentrations for the sum of four furandiones and their $F_P$ are shown in Fig. 6, with
individual species data in Fig. S4. The total furandiones concentration detected in the particle phase ranged from 0.3 to 4.3 ng m⁻³, and averaged 1.6 ± 1.1 ng m⁻³. These concentrations were lower than those detected in our previous study (9.3 ± 3.0 ng m⁻³) (Al-Naiema et al., in review), which is likely due to the rainy and foggy weather in the fall of 2015. In the presence of water, anhydrides undergo hydration and ring opening to form the carboxylic acid derivatives. The relative rate of hydrolysis for FD and, MFD are 6 times higher than DFD and DMFD (Trivedi and Culbertson, 1982). The highest stability
against water hydrolysis might explain the higher concentration of DFD detected in this study compared to other furandiones. The sum of the gas phase concentration for the furandiones (DFD, MFD, and DMFD) averaged 18.0 ± 10.7 ng m⁻³.

Furandiones were almost entirely in the gas phase (Fig. S4). The measured $F_p$ were 0.31 for DFD, 0.08 for MFD, and 0.05 for DMFD, while this value is not reported for FD which showed poor extraction recovery (<10%) from the PUF.
Low $F_P$ values are expected for furandiones due to their high vapour pressure (Table S1). The measured $F_P$ values were substantially higher than those predicted by Pankow's absorption model by two orders of magnitude (Table 2 and Table S1). It is possible that higher than predicted $F_P$ values were driven by furandione adsorption on the front filter or breakthrough from the PUF (Chuang et al., 1987).

Although no sources other than photooxidation of anthropogenic VOCs are known to influence the atmospheric
concentration of furandiones (Forstner et al., 1997; Hamilton et al., 2005), only a moderate correlation ($r_s$= 0.50, p = 0.064) was observed between the particle concentrations of furandiones with DHOPA (Table S3). This may be due to the fact that DHOPA is a tracer specific to toluene, while furandiones can also form from other aromatic VOC (Forstner et al., 1997). Overall, we conclude that furandiones hold a significant importance to serve as indicators for atmospherically-processed aromatic VOC due to their source specificity, however the substantial partitioning toward the gas phase and their water
sensitivity limit their application as SOA tracers.

**4 Conclusions**

This study evaluates, for the first time, the influence of the source specificity, ambient concentration, and gas-particle partitioning on the efficacy of the use of nitromonoaromatics, benzene dicarboxylic acids, furandiones, and DHOPA as tracers for SOA from anthropogenic VOC. First and foremost, DHOPA was detected consistently and only in particle
phase only and is specific to toluene photooxidation, making it a good tracer for toluene SOA despite its relatively low concentrations. Second, PhA is the most abundant benzene dicarboxylic acid isomer and correlates highly with DHOPA.





Similarly, 4M-PhA correlates highly with DHOPA. Although the measured $F_P$ values suggest partitioning to the gas phase for these two species, this is likely due to instrumental interferences from the corresponding anhydrides. Their particle phase concentration, nonetheless, are expected to be useful in tracing naphthalene-derived SOA. Third, 4H-3NBA was detected in only in the particle phase and found to be specific to toluene photooxidation at low levels of $NO_x$. Because of their unique

sources, detectability, and partitioning towards the particle phase, these species are expected to provide much needed insight to SOA from anthropogenic origins, which can support a better understanding of the sources of atmospheric aerosols.

While the above-described species are proposed as tracers of anthropogenic SOA, structurally similar compounds are largely associated with primary sources and are not suitable tracers of SOA. For example, t-PhA, 4NC, 4M-5NC, and 5NSA were highly correlate with levoglucosan and known to be a biomass burning products. Other species such as

furandiones hold significant potential to be used as an indicator of processed aromatic VOC in the atmosphere due to their source specificity, but are not recommended as SOA tracers because of their substantial partitioning in the gas phase and water sensitivity. These findings underscore the importance of evaluating and quantifying potential SOA tracers on an individual species level, as some species within a compound class may provide source specificity while others do not.

**Acknowledgements**

This work was supported by the National Science Foundation (NSF) through AGS grant number 1405014. We thank Carter Madler and Candice Smith for their assistance in the laboratory preparations, and Md. Robiul Islam for help with sample collection.

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



Figure Captions

Figure 1: Fraction of PAHs in the particle phase as a function of the number of carbon atoms. Circles represent the daily measured $F_p$ (n=14), the red line represents average $F_p$, and the dotted line (black) is the predicted $F_p$ using the absorption model (Table S1). Select PAHs were used compare ambient measurements and model estimation: $C_{10}$ is naphthalene, $C_{12}$ is acenaphthene, $C_{14}$ is anthracene, $C_{16}$ is pyrene, $C_{18}$ is benzo(ghi)fluoranthene, $C_{20}$ is benzo(b)fluoranthene, and $C_{22}$ is picene.

Figure 2: Mass spectrum of the trimethylsilylateted deravitives of 2,3-dihydroxy-4-oxopentanoic acid by electron impact ionization.

Figure 3: Daily $PM_{2.5}$ concentrations for 2,3-dihydroxy-4-oxopentanoic acid (a) and benzene dicarboxylic acids (b), where PhA is phthalic acid, t-PhA is terephthalic acid, i-PhA is isophthalic acid, and 4M-PhA is 4-methylphthalic acid.

Figure 4: Selected ion chromatogram for a mixture of nitromonoaromatics standard (5 ng mL$^{-1}$). Where 4M-2NP is 4-methyl-2-nitrophenol, 4NP is 4-nitrophenol, 4NP (IS) is 4-nitrophenol-D4, 4M-3NP is 4-methyl-3-nitrophenol, 2M-4-NP is 2-methyl-4-nitrophenol, 4NG is 4-nitroguaiacol, 5-NG is 5-nitroguaiacol, 4NC is 4-nitrocatechol, 4M-5NC is 4-methyl-5-nitrocatechol, 3NSA (IS) is 3-nitrosalicylic acid-D3, 4H-3NB is 4-Hydroxy-3-nitrobenzyl alcohol, and 5NSA is 5-nitrosalicylic acid.

Figure 5: Time series of ambient $PM_{2.5}$ concentration of levoglucosan, 4-nitrocatechol (4-NC), 4-methyl-5-nitrocatechol (4-M-5NC), and 5-nitrosalicylic acid (5NSA).

Figure 6: Time series of the measured furandiones (2,5-furandione (FD), dihydro-2,5-furandione (DFD), 3-methyl-2,5-furandione (MFD), and dihydro-3-methyl-2,5-furandione (DMFD)) detected in gas and particle phases, with the measured fraction in the particle phase ($F_p$). Furandiones were not detected in the gas phase on 4 November, 2015. Due to poor extraction recoveries, the gas phase concentration of FD was not reported.



**Table 1**: Method performance parameters for nitromonoaromatics compounds, including GC retention time ($t_R$), instrument detection limit (IDL), instrument quantitation limit (IQL), and mean extraction recoveries (± 1 standard deviation for n = 3).

| Nitromonoaromatic | $t_R$ (min) | Base peak[a] (m/z) | Linear range (µg L⁻¹) | Linear regression R² | IDL (µg L⁻¹) | IQL (µg L⁻¹) | Extraction recovery Filter (%) | PUF (%) |
|---|---|---|---|---|---|---|---|---|
| 4-Nitrophenol-D4 (IS) (4NP-D4) | 9.66 | 200 | - | - | - | - | - | - |
| 4-Nitrophenol (4NP) | 9.69 | 196 | 50-5000 | 0.999 | 13.2 | 43.9 | 100.5 ± 2.2 | 97.9 ± 2.2 |
| 4-Methyl-2-nitrophenol (4M-2NP) | 9.71 | 210 | 10-5000 | 0.999 | 2.7 | 8.8 | 96.7 ± 2.8 | 100.3 ± 0.6 |
| 4-Methyl-3-nitrophenol (4M-3NP) | 9.71 | 208 | 40-5000 | 0.999 | 11.4 | 38.0 | 100.2 ± 0.5 | 97.1 ± 3.2 |
| 2-Methyl-4-nitrophenol (2M-4-NP) | 11.08 | 210 | 50-5000 | 0.999 | 14.5 | 48.2 | 99.7 ± 0.8 | 99.0 ± 1.6 |
| 4-Nitroguaiacol (4NG) | 12.08 | 211 | 50-5000 | 0.999 | 14.9 | 49.5 | 94.2 ± 2.4 | 97.9 ± 2.2 |
| 5-Nitroguaiacol (5NG) | 12.40 | 211 | 30-5000 | 0.998 | 6.7 | 22.4 | 96.2 ± 3.3 | 97.4 ± 1.7 |
| 4-Nitrocatechol (4NC) | 13.13 | 284 | 40-5000 | 0.999 | 10.1 | 33.8 | 102.5 ± 2.9 | 49.9 ± 17.0 |
| 4-Methyl-5-nitrocatechol (4M-5NC) | 13.66 | 313 | 40-5000 | 0.999 | 9.1 | 30.3 | 108.1 ± 1.8 | 41.5 ± 24.8 |
| 4-Hydroxy-3-nitrobenzyl alcohol (4H-3NB) | 14.23 | 298 | 20-5000 | 0.998 | 6.3 | 21.0 | 95.6 ± 2.5 | 103.3 ± 3.5 |
| 3-Nitrosalicylic acid-D3 (IS) (3NSA-D3) | 14.08 | 315 | - | - | - | - | - | - |
| 5-Nitrosalicylic acid (5NSA) | 15.11 | 312 | 20-5000 | 0.996 | 5.1 | 16.9 | 100.2 ± 1.0 | 95.0 ± 3.1 |

[a] m/z are given for **trimethylsilylated** derivatives of all analytes





**Table 2:** Summary table for the ambient measured concentrations of organic species in gas and particle phases, measured fraction in the particle phase ($F_p$), frequency of detection in the particle phase ($FOD_p$), and sources reported in the literature.

| Compound | Ambient concentration (ng m$^{-3}$) Mean ($\pm$SD) Particle | Gas | $F_p$ (%) | $FOD_p$ (%) | Some of the reported emission sources in the literature — Secondary photooxidation | Biomass burning | Vehicle emissions |
|---|---|---|---|---|---|---|---|
| 2,3-Dihydroxy-4-oxopentanoic acid | 0.29 (0.12) | ND | 100 | 100 | Kleindienst et al. (2004) | | |
| Phthalic acid | 3.42 (1.92) | 9.62 (3.70) | 26 | 100 | (Kleindienst et al., 2012) | | (Kawamura and Kaplan, 1987) |
| Terephthalic acid | 0.90 (0.58) | ND | 100 | 100 | | | |
| Isophthalic acid | 6.21 (4.82) | ND | 100 | 100 | (Kleindienst et al., 2012) | | |
| 4-Methylphthalic acid | 1.08 (0.51) | 0.23 (0.22) | 82 | 100 | (Kleindienst et al., 2012) | | (Kawamura and Kaplan, 1987) |
| 4-Nitrophenol | 0.63 (0.48) | 1.47 (1.95) | 30 | 100 | (Forstner et al., 1997) | | (Tremp et al., 1993) |
| 4-Methyl-2-nitrophenol | 0.26 (0.09) | 5.13 (8.57) | 5 | 100 | (Forstner et al., 1997) | | (Tremp et al., 1993) |
| 2-Methyl-4-nitrophenol | 0.08 (0.05) | 0.16 (0.15) | 33 | 93 | (Forstner et al., 1997) | | (Tremp et al., 1993) |
| 4-Nitroguaiacol | 0.08 (0.02) | 0.66 (0.76) | 11 | 86 | | (Iinuma et al., 2007) | |
| 4-Nitrocatechol | 1.60 (2.88) | 0.09 (0.07) | 95 | 93 | (Lin et al., 2015) | (Iinuma et al., 2007) | |
| 4-Methyl-5-nitrocatechol | 1.61 (1.77) | 0.08 (0.06) | 95 | 86 | (Lin et al., 2015) | (Iinuma et al., 2007) | |
| 4-Hydroxy-3-nitrobenzyl alcohol | 0.06 (0.06) | ND | 100 | 71 | (Hamilton et al., 2005) | | |
| 5-Nitrosalicylic acid | 0.14 (0.08) | 0.04 (0.03) | 78 | 100 | (Jang and Kamens, 2001) | (Kitanovski et al., 2012) | |
| 2,5-Furandione | 0.60 (0.58) | NR | NR | 36 | (Forstner et al., 1997) | | |
| Dihydro-2,5-furandione | 1.57 (1.34) | 5.71 (3.33) | 0.16 | 100 | (Forstner et al., 1997) | | |
| 3-Methyl-2,5-furandione | 0.44 (0.67) | 7.19 (4.55) | 0.03 | 79 | (Forstner et al., 1997) | | |
| Dihydro-3-methyl-2,5-furandione | 0.63 (0.97) | 5.10 (3.99) | 0.02 | 71 | (Hamilton et al., 2005) | | |
| 2-Methylglyceric acid | 0.68 (0.80) | 0.12 (0.08) | 85 | 100 | (Claeys et al., 2004b) | | |
| 2-Methylthreitol | 9.90 (12.16) | 5.82 (3.89) | 63 | 100 | (Claeys et al., 2004a) | | |
| 2-Methylerythritol | 12.07 (15.51) | 7.20 (4.74) | 63 | 100 | (Claeys et al., 2004a) | | |
| cis-Pinonic acid | 2.56 (3.11) | 1.57 (1.80) | 62 | 100 | (Yu et al., 1999) | | |
| Levoglucosan | 109.68 (68.12) | ND | 100 | 100 | | (Simoneit et al., 1999) | |

ND   Not detected
NR   Not reported (see the text)





Figure 1

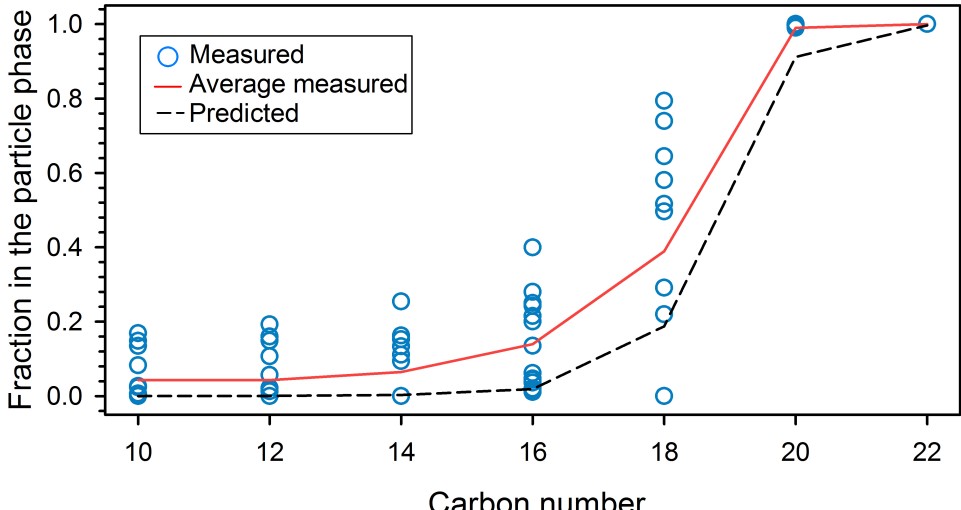



Figure 2

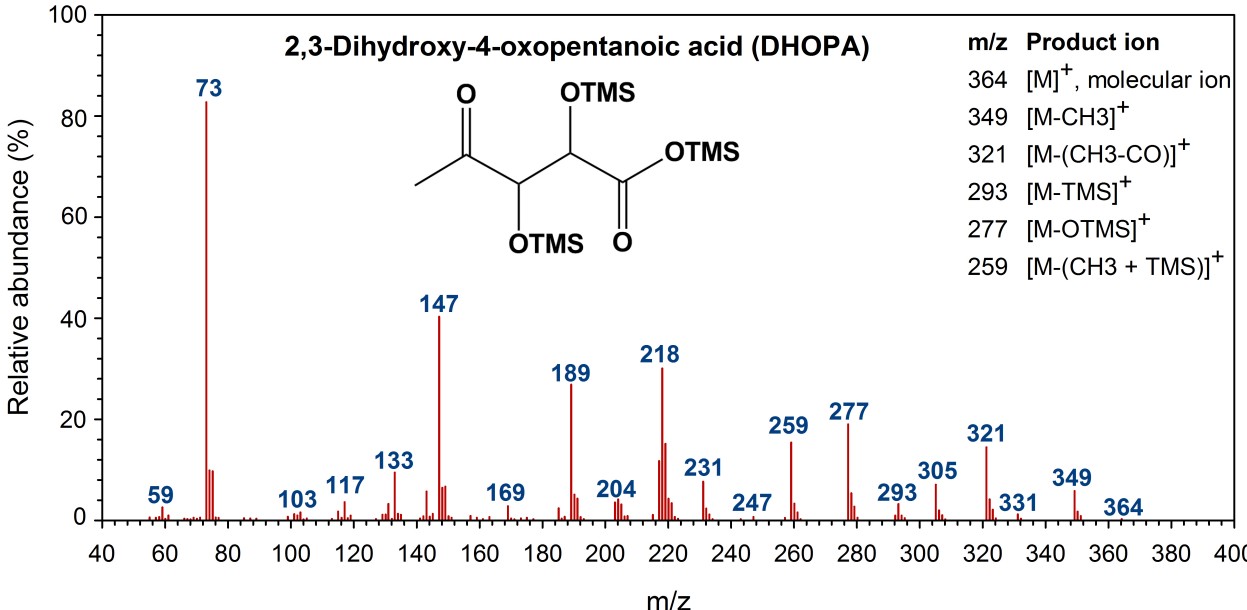





Figure 3

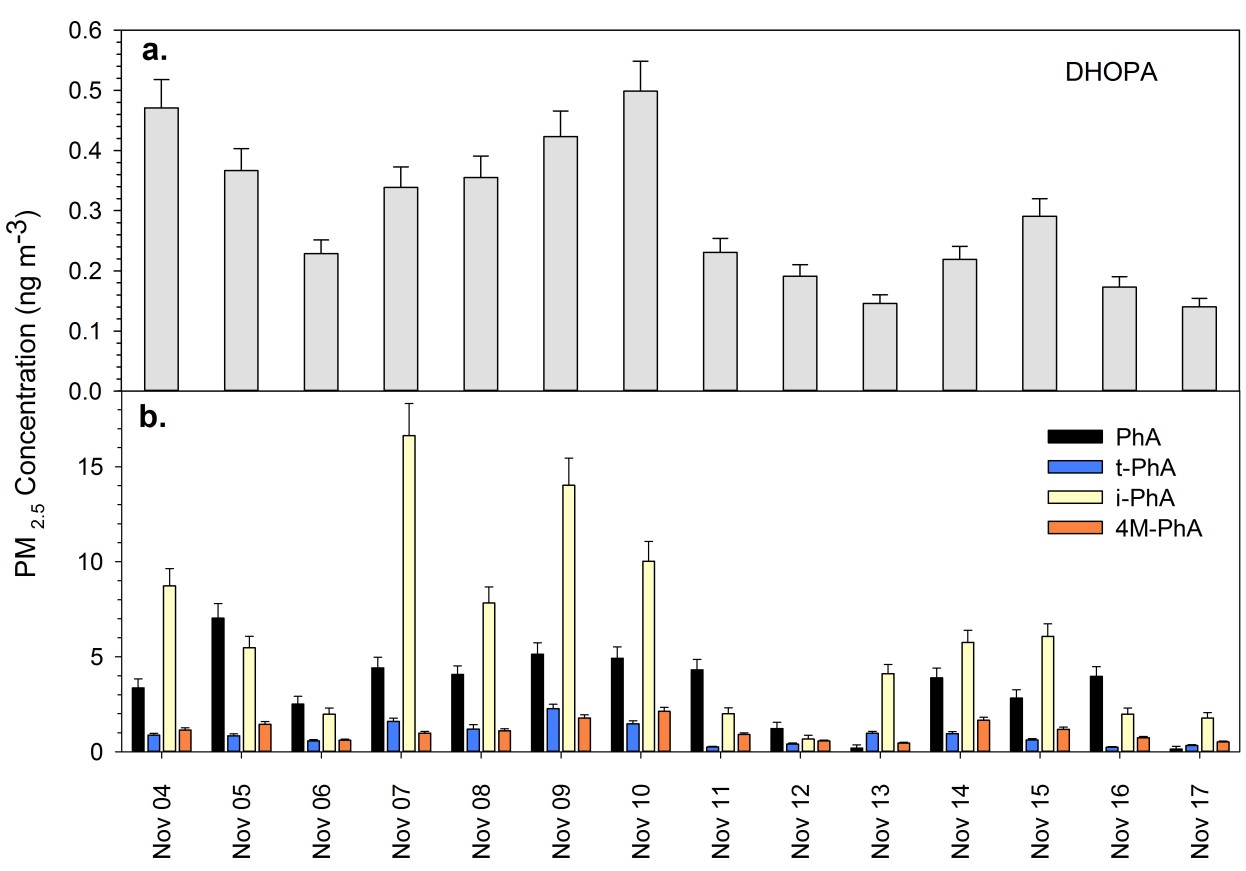



Figure 4

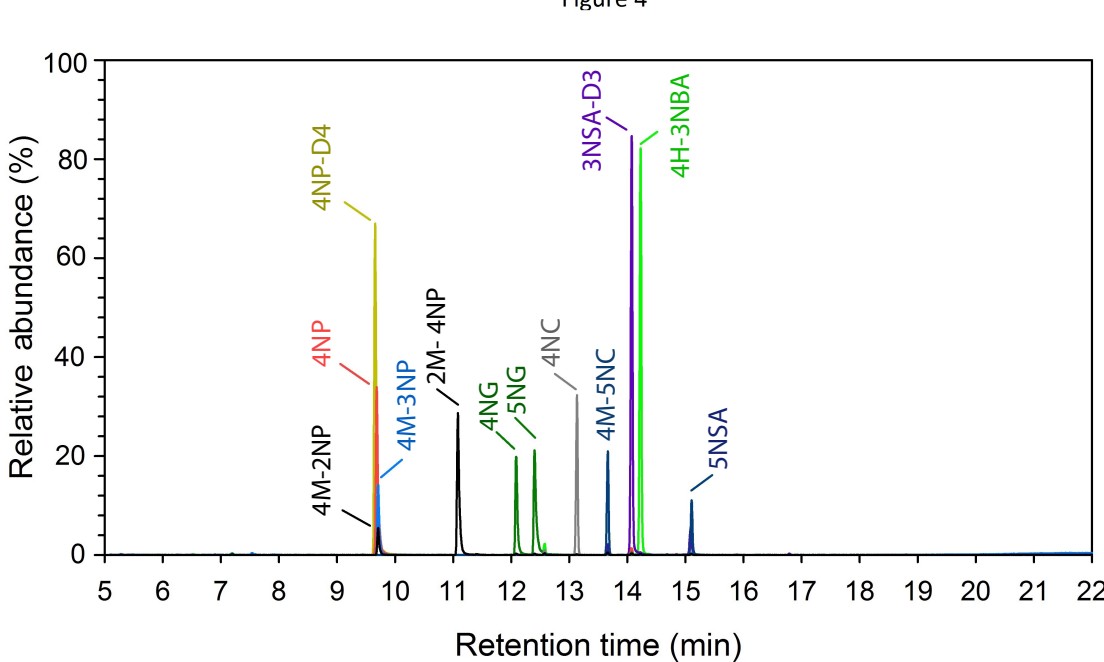



Figure 5

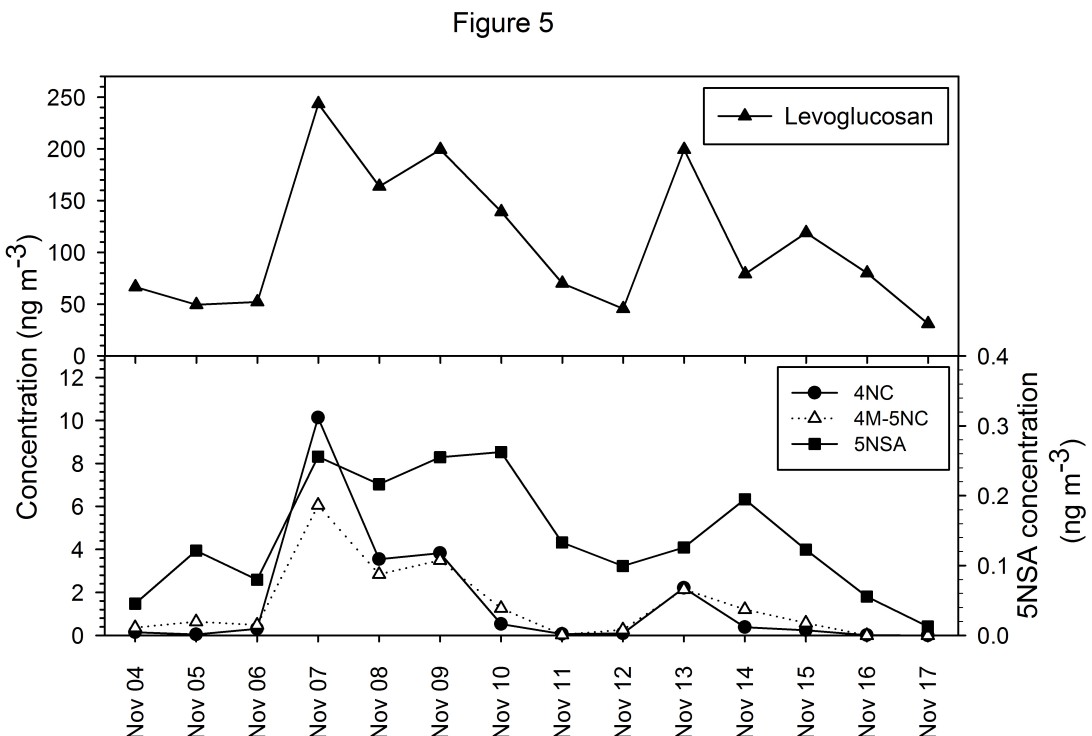



Figure 6

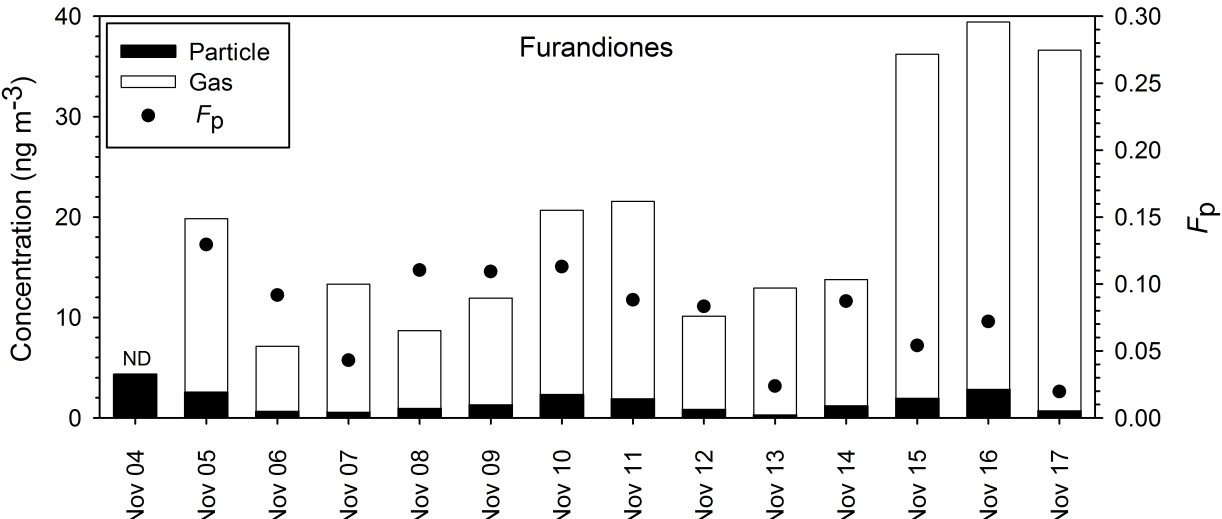