# Peer review of "Evaluation of Anthropogenic Secondary Organic Aerosol Tracers from Aromatic Hydrocarbons"

_Atmospheric Chemistry and Physics, 2016_

## Referee Comment (RC1) · Anonymous Referee #1 · 26 Sep 2016

General comments: The authors selected 2,3-dihydroxy-4-oxopentanoic acid (DHOPA), phthalic acid derivatives, nitroaromatic compounds, and frandiones as the candidates of anthropogenic secondary organic aerosol (ASOA) tracer to test them based on a field observation campaign. Secondary organic aerosol is believed to affect climate, visibility, and human health. A tracer-based approach is hopeful technique to identify aerosol sources. Currently DHOPA is an only established tracer of ASOA, therefore additional ASOA tracers would be helpful for better understanding. The authors collected gas and particle offline samples in November, 2015 at a site in Iowa City, United States. The authors analyzed sample extracts employing TMS-derivatization gas chromatography-mass spectrometry. The authors studied source specificity, con-

sistent detactability, atmospheric stability, and partitioning to the aerosol phase for each candidate to conclude that DHOPA, phthalic acid, and 4-hydroxy-3-nitrobenzyl alcohol will be used as ASOA tracers. The manuscript is well written and will provide new physical insight into the atmospheric chemistry, but the following comments should be addressed before publication:

Comments:

(1) Page 3, line 22. Is November the suitable season to study the validity of ASOA markers? Discussion on observation period would be necessary.

(2) Page 4, line 14. Ketopinic acid would have lower polarity than DHOPA. 13C-Labeled adipic acid or deuterated tartaric acid might be better internal standards for DHOPA. Discussion on the internal standard of DHOPA would be necessary.

(3) Page 4, lines 17-18. The power of sonication is missing.

(4) Page 8, lines 31. Is 4M-PhA the product of naphthalene photooxidation? It would be produced by the reaction of an isomer of methylnaphthalene.

(5) Page 9, lines 8-10. The definition of M is missing. If M indicates the molecular mass of derivative, [M-NO2-CH3]+ and [M-NO2-CH3-Si(CH3)3]+ would be measured at m/z of M-61 and M-134, respectively.

(6) Page9, line 10. [M-NO2]+ fragment would be measured at m/z of M-46 if M is the molecular mass of derivative. According to Table S2, [M-NO2]+ fragment would be observed for methylnitrophenol isomers rather than nitroguaiacol isomers.

(7) Page 24, Figure 4. The chromatographic peaks of 4NP and 4M-2NP are overlapped. These fragment signals might be interfered each other. Discussion on these interferences would be necessary in text.

---

## Referee Comment (RC2) · Anonymous Referee #2 · 18 Oct 2016

This paper describes analytical methods and a test ambient study to evaluate the applicability of a range of compounds to be used as tracers for SOA produced from anthropogenic aromatic VOCs.

I offer the following comments and suggestions to help improve the manuscript.

1. Abstract: The title's focus of "anthropogenic" SOA tracers is focused away from "anthropogenic" and toward "aromatic" derived SOA tracers in the abstract. Maybe "aromatic" should also be included somehow in the title.

2. Page 4, Line 32: A different derivatization condition was used for Nitromonoaromatics, but the reasoning for a different method is not discussed. Please describe why

alternate method was applied.

3. Page 5, Line 4: update Al-Naiema et al., in review if possible

4. Page 5, Line 24: assuming activity coefficient =1 introduces some large potential errors. Knowing something about the classes of interest and the greater aerosol mixture could help justify this assumption, or help to assume an alternate activity coefficient. Optionally, a lower and higher value could be incorporated in the final table to show a range of resulting partitioning values.

5. Page 7, Line 17: Regarding, "DHOPA accounted for $0.0079 \pm 0.0026$ of secondary OC mass", is that a fraction or percentage?

6. Page 7, Line 18: SOC has not been defined previously

7. Page 11, Line 9: Be sure to define all abbreviations before use in main text (e.g. FD and MFD here).

8. General: Would be useful to have another figure, maybe in Supplement, that incorporates other daily metrics from Nov4-17 such as T, RH, OC, EC, PM2.5, and any other supporting info.

9. General: With an understanding of the amount of work devoted to developing and testing new methods, the ambient results would be much more meaningful with a longer sample period, and a contrast with different seasons or different locations. However, I think the results provided on new methods and a test-case ambient example are sufficient to highlight good candidate tracers, but solid conclusions should be reserved for longer sample periods, various seasons (especially summer), and additional environments.

---

## Author Comment (AC1) · 21 Dec 2016

Anonymous referee #1 general comments: "The authors selected 2,3-dihydroxy-4-oxopentanoic acid (DHOPA), phthalic acid derivatives, nitroaromatic compounds, and furandiones as the candidates of anthropogenic secondary organic aerosol (ASOA) tracer to test them based on a field observation campaign. Secondary organic aerosol is believed to affect climate, visibility, and human health. A tracer-based approach is hopeful technique to identify aerosol sources. Currently DHOPA is an only established tracer of ASOA, therefore additional ASOA tracers would be helpful for better understanding. The authors collected gas and particle offline samples in November, 2015 at a site in Iowa City, United States. The authors analyzed sample extracts em-

ploying TMS-derivatization gas chromatography-mass spectrometry. The authors studied source specificity, consistent detectability, atmospheric stability, and partitioning to the aerosol phase for each candidate to conclude that DHOPA, phthalic acid, and 4-hydroxy-3-nitrobenzyl alcohol will be used as ASOA tracers. The manuscript is well written and will provide new physical insight into the atmospheric chemistry, but the following comments should be addressed before publication"

Response to referee #1 general comments: We agree with the referee's summary of our manuscript. We thank the referee for the valuable comments regarding the manuscript. Specific comments are addressed point-by-point below.

Referee #1 comment 1 – Page 3, line 22: "Is November the suitable season to study the validity of ASOA markers? Discussion on observation period would be necessary."

Response to referee #1 comment 1: We agree that it is important to justify the timing of this field study. To clarify this, the following text has been added to the introduction page 3, line 15: "November was chosen for this study because, in a prior study at this site, biogenic SOA tracers were detected in this month (Jayarathne et al., 2016) and aromatic SOA tracers have a less pronounced seasonal variation than those that are biogenic (Shen et al., 2015; Ding et al., 2012; Lewandowski et al., 2008).

Referee #1 comment 2 – Page 4, line 14: "Ketopinic acid would have lower polarity than DHOPA. 13C-Labeled adipic acid or deuterated tartaric acid might be better internal standards for DHOPA. Discussion on the internal standard of DHOPA would be necessary."

Response to referee #1 comment 2: It is important that the internal standard match the analyte in physical properties, such as solubility, volatility, and reactivity, so that the internal standard can properly normalize out any deviations arising during the extraction, GC analysis, and derivatization process, respectively (Zhang et al., 2009). Ketopinic acid was selected as an internal standard, because of the precedence set by founding studies of the SOA tracer method (Kleindienst et al., 2007; Kleindienst et al.,

2009). Using a DHOPA standard and ketopinic acid internal standard, we show that the extraction efficiency for DHOPA was $98.7 \pm 1.8\%$ (n=3) and thus demonstrate the effectiveness and suitability of ketopinic acid as an internal standard for DHOPA. To clarify this in the text, the following text is added to page 4 line 17: "The use of KPA as internal standard for DHOPA builds upon prior work by Kleindienst et al. (2007)."

Referee #1 comment 3 – Page 4, line 17-18: "The power of sonication is missing."

Response to referee #1 comment 3: The sonicator we use has a power of 137 W. This information has been added at page 4 line 19: "Filters were extracted sequentially with three 10 mL portions of acetonitrile using ultrasonication (Branson 5510, 137 W) for 15 minutes at 60 sonics per minute."

Referee #1 comment 4 – Page 8, line 31: "Is 4M-PhA the product of naphthalene photooxidation? It would be produced by the reaction of an isomer of methylnaphthalene."

Response to referee #1 comment 4: We thank the referee for bringing this to our attention. There is no evidence that 4M-PhA would produce from naphthalene; instead it is produced from methylnaphthalene.

The following text has been removed from page 8 line 30: "Together, the relative high concentration detected in the particle phase relative to other tracers, and the high correlations with DHOPA suggest that PhA and 4M-PhA are useful SOA tracers for naphthalene photooxidation." The new text reads: "Together, the relative high concentration detected in the particle phase relative to other tracers, and the strong correlations with DHOPA, suggesting PhA and 4M-PhA as useful SOA tracers for naphthalene and methylnaphthalene photooxidation, respectively." Referee #1 comment 5 - Page 9, line 8-10: "The definition of M is missing. If M indicates the molecular mass of derivative, [M-NO2-CH3]+ and [M-NO2-CH3-Si(CH3)3]+ would be measured at m/z of M-61 and M-134, respectively."

Response to referee #1 comment 5: We thank the referee for bring this point to our

attention. There were some typos with the reported numbers, and M refers to the molecular ion for the trimethylsilylated ester.

The following text has been removed from page 9 line 8-10: "Save for nitroguaiacols, nitromonoaromatics mass spectra included mass fragments of [M-57]+ (loss of NO2 and CH3) and [M-129]+ (loss of (Si(CH3)3) for the singly and doubly derivatized analytes, respectively. Nitroguaiacol isomers had a fragment at [M-42]+ 10 (loss of NO2).."

The new text reads: Nitromonoaromatic mass spectra (Table S2) included mass fragments with m/z [M-60]+ (from the loss of NO2 and CH3), where M is molecular ion for the trimethylsilylated ester. Save for nitroguaiacols and 4-methyl-5-nitrocatechol, other nitromonoaromatics mass spectra included a mass fragment of [M-15]+ (loss of CH3).

Referee #1 comment 6 - Page 9, line 10: "[M-NO2]+ fragment would be measured at m/z of M-46 if M is the molecular mass of derivative. According to Table S2, [M-NO2]+ fragment would be observed for methylnitrophenol isomers rather than nitroguaiacol isomers."

Response to referee #1 comment 6: We thank the referee for pointing out this typographical error. This error has been corrected in our response to comment 5.

Referee #1 comment 7 - Page 24, Figure 4: "The chromatographic peaks of 4NP and 4M-2NP are overlapped. These fragment signals might be interfered each other. Discussion on these interferences would be necessary in text.

Response to referee #1 comment 7: We thank the referee for pointing this out. To clarify that we have investigated the possible interferences from the co-eluting nitromonoaromatic peaks, the following text has been added at page 9 line 19: "The mass spectra for the co-eluting peaks (Figure 4, Table S2) indicates that potential interferences for the 4NP-D4, 4NP, and 4M-2NP are not appreciably strong (< 1%), and thus interferences are expected to be negligible. There is potential for 4M-3NP to interfere with detection of 4M-2NP, because the former shows a relatively strong signal for m/z 210 (at 38%

of the base peak signal) that is used to quantify the latter; however 4M-3NP was not detected in this study, so no interference is expected.

Works Cited

Ding, X., X.-M. Wang, B. Gao, X.-X. Fu, Q.-F. He, X.-Y. Zhao, J.-Z. Yu and M. Zheng, 2012. Tracer-based estimation of secondary organic carbon in the Pearl River Delta, south China. Journal of Geophysical Research-Atmospheres 117.

Jayarathne, T., C. M. Rathnayake and E. A. Stone, 2016. Local source impacts on primary and secondary aerosols in the Midwestern United States. Atmospheric Environment 130, 74-83.

Kleindienst, T. E., M. Jaoui, M. Lewandowski, J. H. Offenberg, C. W. Lewis, P. V. Bhave and E. O. Edney, 2007. Estimates of the contributions of biogenic and anthropogenic hydrocarbons to secondary organic aerosol at a southeastern US location. Atmospheric Environment 41 (37), 8288-8300.

Kleindienst, T. E., M. Lewandowski, J. H. Offenberg, M. Jaoui and E. O. Edney, 2009. The formation of secondary organic aerosol from the isoprene plus OH reaction in the absence of NOx. Atmospheric Chemistry and Physics 9 (17), 6541-6558.

Lewandowski, M., M. Jaoui, J. H. Offenberg, T. E. Kleindienst, E. O. Edney, R. J. Sheesley and J. J. Schauer, 2008. Primary and secondary contributions to ambient PM in the midwestern United States. Environmental Science & Technology 42 (9), 3303-3309.

Shen, R. Q., X. Ding, Q. F. He, Z. Y. Cong, Q. Q. Yu and X. M. Wang, 2015. Seasonal variation of secondary organic aerosol tracers in Central Tibetan Plateau. Atmospheric Chemistry and Physics 15 (15), 8781-8793.

Zhang, Y. X., J. J. Schauer, E. A. Stone, Y. H. Zhang, M. Shao, Y. J. Wei and X. L. Zhu, 2009. Harmonizing Molecular Marker Analyses of Organic Aerosols. Aerosol Science and Technology 43 (4), 275-283.

---

## Author Comment (AC2) · 21 Dec 2016

Anonymous referee #2 general comments: "This paper describes analytical methods and a test ambient study to evaluate the applicability of a range of compounds to be used as tracers for SOA produced from anthropogenic aromatic VOCs. I offer the following comments and suggestions to help improve the manuscript."

Response to referee #2 general comments: We thank the referee for the careful review and suggestions to improve the manuscript. We address the specific questions and comments point-by-point below.

Referee #2 comment 1 – Abstract: "The title's focus of "anthropogenic" SOA tracers is

focused away from "anthropogenic" and toward "aromatic" derived SOA tracers in the abstract. Maybe "aromatic" should also be included somehow in the title."

Response to referee #2 comment 1: As suggested by the referee, the title for this paper has been revised to reflect the aromatic origin of the investigated tracers. The new title of the manuscript is: "Evaluation of Anthropogenic Secondary Organic Aerosol Tracers from Aromatic Hydrocarbons"

Referee #2 comment 2 – Page 4, line 32: "different derivatization condition was used for nitromonoaromatics, but the reasoning for a different method is not discussed. Please describe why alternate method was applied."

Response to referee #2 comment 2: To clarify why we used a different derivatization conditions for nitromonoaromatics the following text has been added to Page 5 line 5: "The different silylation protocol used for nitromonoaromatics yielded more symmetrical peak shapes and higher intensities, compared to the derivatization method used for levoglucosan and phthalic acid isomers that resulted in asymmetrical nitromonoaromatic peaks with low intensities." Referee #2 comment 3 – Page 5, line 4: "update Al-Naiema et al., in review if possible."

Response to referee #2 comment 3: We thank the referee for pointing out this point. This manuscript is currently in revision at Atmospheric Environment and we anticipate being able to update this soon.

Referee #2 comment 4 – Page 5, line 24: "assuming activity coefficient =1 introduces some large potential errors. Knowing something about the classes of interest and the greater aerosol mixture could help justify this assumption, or help to assume an alternate activity coefficient. Optionally, a lower and higher value could be incorporated in the final table to show a range of resulting partitioning values."

Response to referee #2 comment 4: We thank the referee for his valuable suggestion. Activity coefficient (zeta) for atmospheric organic compounds in the literature ranges

from 0.3-3.0 (Seinfeld and Pankow, 2003). We followed the referee's suggestion and included the upper and lower Fp values for the compounds of interest. A new column has been added to Table S1 in the supplement. As indicated in Table S1, the calculated fractions in the particle phase show no significant changes in FP for phthalic acid isomers and furandiones when moving from an activity coefficient of 0.3 to 3.

The following text has been removed from the Supplement: "Table S1: Parameters used to calculate gas-particle partitioning by absorption theory. The activity coefficients were assumed to equal one, subcooled vapour pressures were obtained from the Estimation Programs Interface suite (EPA, 2012). The vapour pressure values were corrected for average ambient temperature during sample collection (282 K) using Clausius-Clapeyron equation, then the corrected values were used to calculate the partitioning coefficient, expressed as fraction in the particle phase (FP).

The new text reads: "Table S1: Parameters used to calculate gas-particle partitioning by absorption theory at three different values of activity coefficients (zeta), each of 0.3, 1, and 3 (Seinfeld and Pankow, 2003), representing the upper, middle, and lower range of calculated fraction in the particle phase (FP), respectively. Subcooled vapour pressures were obtained from the Estimation Programs Interface suite (EPA, 2012). The vapour pressure values were corrected for average ambient temperature during sample collection (282 K) using Clausius-Clapeyron equation, then the corrected values were used to calculate the partitioning coefficient, expressed FP."

Referee #2 comment 5 - Page 7, line 17: "Regarding, "DHOPA accounted for 0.0079 ± 0.0026 of secondary OC mass", is that a fraction or percentage?

Response to referee #2 comment 5: We thank the referee for bring this point to our attention. These numbers represent the mass fraction of DHOPA to the total OC formed from toluene photooxidation. The following text has been removed from Page 7, line 17: "The contribution of toluene SOA to OC was estimated based on the SOA-tracer method introduced by Kleindienst et al., (2007), where DHOPA accounted for 0.0079

± 0.0026 of secondary OC mass from toluene." The new text reads: "The contribution of toluene SOA to OC was estimated based on the SOA-tracer method introduced by Kleindienst et al., (2007), where the DHOPA mass fraction of secondary organic carbon (SOC) from toluene was 0.0079 ± 0.0026."

Referee #2 comment 6 - Page 7, line 18: "SOC has not been defined previously."

Response to referee #2 comment 6: We thank the referee for bringing this to our attention to this point. SOC is now defined at page 7 on line 23, with the revised text provided in response to referee #2 comment 5.

Referee #2 comment 7 - Page 11, Line 9: "Be sure to define all abbreviations before use in main text (e.g. FD and MFD here)."

Response to referee #2 comment 7: We thank the referee for pointing out this important point. The following text has been removed from Page 11, Line 23: "The relative rate of hydrolysis for FD and, MFD are 6 times higher than DFD and DMFD (Trivedi and Culbertson, 1982)."

The new text reads:" The relative rate of hydrolysis for 2,5-furandione (FD) and, 3-methyl-2,5-furandione (MFD) are 6 times higher than dihydro-2,5-furandione (DFD) and dihydro-3-methyl-2,5,-furandione (DMFD) (Trivedi and Culbertson, 1982).

Referee #2 comment 8 - General: "Would be useful to have another figure, maybe in Supplement that incorporates other daily metrics from Nov4-17 such as T, RH, OC, EC, PM2.5, and any other supporting info."

Response to referee #2 comment 8: We thank the referee for his suggestion. While PM2.5 measurements are not available for this study, the rest of the suggested information are included to Table S4 in the supplement.

Referee #2 comment 9 - General: "With an understanding of the amount of work devoted to developing and testing new methods, the ambient results would be much more meaningful with a longer sample period, and a contrast with different seasons

or different locations. However, I think the results provided on new methods and a test-case ambient example are sufficient to highlight good candidate tracers, but solid conclusions should be reserved for longer sample periods, various seasons (especially summer), and additional environments."

Response to referee #2 comment 9: We thank the reviewer and agree that the number of the tested samples provide some limitations, taking the seasonal and diurnal variations of the SOA tracers. We are in the process of testing these tracers using longer time period, and in different seasons, time of the day, and locations to understand the value of applying these compounds to trace anthropogenic SOA in different environments.

The following text has been added to the conclusion, Line 28: "Given the limited time and geographic distribution for the samples analyzed in this study, further investigation is needed to realize the value these compounds as tracers of anthropogenic SOA more broadly."

Works Cited

EPA: Estimation Programs Interface Suite$^{TM}$ for Microsoft$^{®}$ Windows, v 4.11 or insert version used. United States Environmental Protection Agency, Washington, DC, USA., in, 2012.

Seinfeld, J. H. and J. F. Pankow, 2003. Organic atmospheric particulate material. Annual Review of Physical Chemistry 54, 121-140.

Please also note the supplement to this comment:
http://www.atmos-chem-phys-discuss.net/acp-2016-805/acp-2016-805-AC2-supplement.pdf

[Figure]

|  | OC (µg m-3) Average ± SD | EC (µg m-3) Average ± SD | Average daily Temperature (°C) | RH (%) |
|---|---|---|---|---|
| 11/4/2015 | 1.76 ± 0.10 | 0.14 ± 0.02 | 14 | 87 |
| 11/5/2015 | 0.66 ± 0.05 | 0.11 ± 0.02 | 15 | 85 |
| 11/6/2015 | 1.12 ± 0.07 | 0.11 ± 0.02 | 7 | 72 |
| 11/7/2015 | 2.28 ± 0.12 | 0.25 ± 0.02 | 5 | 68 |
| 11/8/2015 | 1.95 ± 0.11 | 0.26 ± 0.02 | 5 | 68 |
| 11/9/2015 | 2.90 ± 0.15 | 0.38 ± 0.03 | 5 | 65 |
| 11/10/2015 | 3.01 ± 0.16 | 0.26 ± 0.02 | 6 | 70 |
| 11/11/2015 | 1.95 ± 0.11 | 0.13 ± 0.02 | 11 | 81 |
| 11/12/2015 | 1.07 ± 0.07 | 0.03 ± 0.02 | 6 | 74 |
| 11/13/2015 | 1.56 ± 0.09 | 0.18 ± 0.02 | 4 | 57 |
| 11/14/2015 | 2.12 ± 0.12 | 0.13 ± 0.02 | 8 | 66 |
| 11/15/2015 | 9.65 ± 0.49 | 0.81 ± 0.04 | 10 | 59 |
| 11/16/2015 | 1.37 ± 0.08 | 0.10 ± 0.02 | 9 | 72 |
| 11/17/2015 | 1.23 ± 0.08 | 0.05 ± 0.02 | 13 | 95 |

Table S4: Daily measurements of organic carbon (OC), elemental carbon (EC), ambient temperature and relative humidity (RH), in Iowa City, IA from Nov 4 - Nov. 17, 2011.

**Fig. 1.**

**Supplement:**

Table S1: Parameters used to calculate gas-particle partitioning by absorption theory at three different values of activity coefficients ($\mathcal{E}$), each of 0.3, 1, and 3 (Seinfeld and Pankow, 2003), representing the upper, middle, and lower range of calculated fraction in the particle phase ($F_P$), respectively. Subcooled vapour pressures were obtained from the Estimation Programs Interface suite (EPA, 2012). The vapour pressure values were corrected for average ambient temperature during sample collection (282 K) using Clausius-Clapeyron equation, then the corrected values were used to calculate the partitioning coefficient, expressed $F_P$.

| Compound | Number of carbon atoms | Molecular mass (g mol$^{-1}$) | Subcooled vapour pressure at 298.15 K (atm) | Enthalpy of vaporization (kJ mol$^{-1}$) | Partitioning coefficient at $\mathcal{E}$ =1 (m$^3$ µg$^{-1}$) | Calculated ($F_p$) | | |
|---|---|---|---|---|---|---|---|---|
| | | | | | | $\mathcal{E}$=0.3 | $\mathcal{E}$=1 | $\mathcal{E}$=3 |
| Naphthalene | 10 | 128.18 | $3.93 \times 10^{-4}$ | 60.3 [a] | $1.82 \times 10^{-06}$ | $1.4 \times 10^{-5}$ | $4.2 \times 10^{-6}$ | $1.4 \times 10^{-6}$ |
| Acenaphthene | 12 | 154.21 | $1.34 \times 10^{-5}$ | 70.5 [a] | $5.62 \times 10^{-5}$ | $4.4 \times 10^{-4}$ | $1.3 \times 10^{-4}$ | $4.4 \times 10^{-5}$ |
| Anthracene | 14 | 178.24 | $6.50 \times 10^{-7}$ | 79.5 [a] | $1.23 \times 10^{-3}$ | $9.5 \times 10^{-3}$ | $2.9 \times 10^{-3}$ | $9.6 \times 10^{-4}$ |
| Pyrene | 16 | 202.26 | $1.05 \times 10^{-7}$ | 87.5 [a] | $8.11 \times 10^{-3}$ | $5.9 \times 10^{-2}$ | $1.9 \times 10^{-2}$ | $6.3 \times 10^{-3}$ |
| Benzo(ghi)fluoranthene | 18 | 226.28 | $4.60 \times 10^{-9}$ | 65.2 [a] | $9.89 \times 10^{-2}$ | 0.43 | 0.19 | 0.07 |
| Benzo(a)pyrene | 20 | 252.32 | $2.28 \times 10^{-10}$ | 105 [a] | 4.45 | 0.97 | 0.91 | 0.78 |
| Picene | 22 | 278.36 | $1.19 \times 10^{-11}$ | 135 [a] | 153.34 | 1.00 | 1.00 | 0.99 |
| Phthalic anhydride | 8 | 148.12 | $7.57 \times 10^{-6}$ | 52.9 [b] | $6.94 \times 10^{-5}$ | $5.4 \times 10^{-4}$ | $1.6 \times 10^{-4}$ | $5.4 \times 10^{-5}$ |
| Phthalic acid | 8 | 166.13 | $8.91 \times 10^{-8}$ | 74.1 [b] | $8.54 \times 10^{-3}$ | $6.2 \times 10^{-2}$ | $2.0 \times 10^{-2}$ | $6.6 \times 10^{-3}$ |
| Isophthalic acid | 8 | 166.13 | $1.61 \times 10^{-7}$ | 84.3 [b] | $5.97 \times 10^{-3}$ | $4.4 \times 10^{-2}$ | $1.4 \times 10^{-2}$ | $4.6 \times 10^{-3}$ |
| Terephthalic acid | 8 | 166.13 | $1.22 \times 10^{-7}$ | 93.4 [b] | $9.67 \times 10^{-3}$ | $7.0 \times 10^{-2}$ | $2.2 \times 10^{-2}$ | $7.5 \times 10^{-3}$ |
| 2,5-Furandione | 4 | 98.06 | $6.19 \times 10^{-4}$ | 56.7 [c] | $3.80 \times 10^{-7}$ | $3.0 \times 10^{-6}$ | $8.8 \times 10^{-7}$ | $3.0 \times 10^{-7}$ |
| Dihydro-2,5-furandione | 4 | 100.07 | $1.15 \times 10^{-5}$ | 57.3 [c] | $2.00 \times 10^{-5}$ | $1.6 \times 10^{-5}$ | $4.6 \times 10^{-5}$ | $1.6 \times 10^{-6}$ |
| 3-Methyl-2,5-furandione | 5 | 112.08 | $2.22 \times 10^{-4}$ | 53.1 [c] | $9.29 \times 10^{-7}$ | $7.2 \times 10^{-6}$ | $2.1 \times 10^{-6}$ | $7.2 \times 10^{-7}$ |
| Dihydro-3-methyl-2,5-furandione | 5 | 114.09 | $4.98 \times 10^{-5}$ | 59.3 [c] | $4.07 \times 10^{-6}$ | $3.2 \times 10^{-5}$ | $9.4 \times 10^{-6}$ | $3.2 \times 10^{-7}$ |

[a] (Kluwer, 1988), [b] (Yaws, 2009), [c] (Linstrom, 2005)

Table S2: Mass spectral fragmentations for trimethylsilylated derivatives of the nitromonoaromatic compounds using electron impact ionization (70 eV). Base peaks were used for quantification, and other fragments were used as qualifiers. Results are listed according to the observed intensities (highest to lowest).

| Nitromonoaromatics | Molecular mass (g mol$^{-1}$) | Calculated der. mass (g mol$^{-1}$) | MS Fragments ($m/z$) | | | | | |
|---|---|---|---|---|---|---|---|---|
| | | | Base peak | 1 | 2 | 3 | 4 | 5 |
| 4-Nitrophenol 2,3,5,6-D$_4$ (IS) | 143.13 | 215.13 | 200 | 215 | 154 | 139 | 73 | 45 |
| 4-Nitrophenol | 139.11 | 211.11 | 196 | 211 | 150 | 135 | 73 | 45 |
| 4-Nitrocatechol | 155.11 | 299.11 | 284 | 299 | 73 | 45 | | |
| 4-Methyl-5-nitrocatechol | 169.13 | 313.13 | 313 | 73 | 296 | 266 | 180 | 45 |
| 4-Nitroguaiacol | 169.13 | 241.13 | 211 | 226 | 241 | 181 | 73 | 45 |
| 5-Nitroguaiacol | 169.13 | 241.13 | 211 | 226 | 241 | 181 | 73 | 45 |
| 4-Methyl-2-nitrophenol | 153.14 | 225.14 | 210 | 165 | 179 | 73 | 225 | 45 |
| 5-Methyl-2-nitrophenol | 153.14 | 225.14 | 210 | 165 | 179 | 73 | 45 | 225 |
| 4-Methyl-3-nitrophenol | 153.14 | 225.14 | 208 | 163 | 225 | 180 | 73 | 45 |
| 2-Methyl-4-nitrophenol | 153.14 | 225.14 | 210 | 225 | 164 | 149 | 73 | 45 |
| 4-Hydroxy-3-nitrobenzyl alcohol | 169.14 | 313.14 | 298 | 224 | 147 | 179 | 313 | 45 |
| 4-Methoxy-2-nitrophenol | 169.13 | 241.13 | 226 | 181 | 241 | 153 | 73 | 45 |
| 3-Nitrosalicylic acid-D$_3$ | 186.14 | 330.14 | 315 | 73 | 147 | 236 | 329 | 45 |
| 5-Nitrosalicylic acid | 183.12 | 327.12 | 312 | 73 | 45 | 326 | | |

Table S3: Spearman's correlation coefficients ($r_s$) for $PM_{2.5}$ tracer concentrations reported in this study (n=14).

| | DHOPA | PhA | i-PhA | t-PhA | 4M-PhA | 4NP | 4M-2NP | 2M-4NP | 4NG | 4NC | 4M-5NC | 4H-3NB | 5NSA | Furandiones [a] | Hopanes [b] | Levoglucosan |
|---|---|---|---|---|---|---|---|---|---|---|---|---|---|---|---|---|
| DHOPA | x | | | | | | | | | | | | | | | |
| PhA | 0.7 | x | | | | | | | | | | | | | | |
| i-PhA | 0.8 | 0.6 | x | | | | | | | | | | | | | |
| t-PhA | 0.6 | 0.5 | 0.9 | x | | | | | | | | | | | | |
| 4M-PhA | 0.8 | 0.7 | 0.7 | 0.5 | x | | | | | | | | | | | |
| 4NP | 0.2 | 0.1 | 0.3 | 0.4 | 0.4 | x | | | | | | | | | | |
| 4M-2NP | 0.1 | 0.3 | 0.2 | 0.4 | 0.4 | 0.2 | x | | | | | | | | | |
| 2M-4NP | 0.4 | 0.1 | 0.3 | 0.4 | 0.5 | 0.8 | 0.4 | x | | | | | | | | |
| 4NG | 0.1 | 0.2 | 0.4 | 0.3 | 0.0 | - 0.1 | 0.1 | - 0.1 | x | | | | | | | |
| 4NC | 0.1 | 0.1 | 0.7 | 0.9 | 0.2 | 0.1 | 0.2 | 0.0 | 0.2 | x | | | | | | |
| 4M-5NC | 0.2 | 0.3 | 0.7 | 0.9 | 0.3 | 0.2 | 0.5 | 0.1 | 0.4 | 0.9 | x | | | | | |
| 4H-3NB | 0.3 | 0.2 | 0.5 | 0.4 | - 0.1 | - 0.2 | 0.2 | 0.1 | 0.5 | 0.4 | 0.5 | x | | | | |
| 5NSA | 0.5 | 0.6 | 0.7 | 0.8 | 0.6 | 0.4 | 0.2 | 0.4 | 0.2 | 0.7 | 0.7 | 0.0 | x | | | |
| Furandiones [a] | 0.5 | 0.5 | 0.2 | - 0.2 | 0.6 | 0.0 | - 0.2 | 0.0 | 0.1 | - 0.5 | - 0.4 | - 0.1 | - 0.2 | x | | |
| Hopanes [b] | 0.2 | 0.2 | 0.6 | 0.8 | 0.2 | 0.3 | 0.3 | 0.2 | 0.1 | 0.9 | 0.8 | 0.1 | 0.9 | - 0.6 | x | |
| Levoglucosan | 0.3 | 0.4 | 0.7 | 0.7 | 0.3 | 0.2 | 0.0 | - 0.1 | 0.4 | 0.9 | 0.9 | 0.2 | 0.8 | - 0.2 | 0.8 | x |

Legend:
- $p - value \leq 0.001$
- $p - value$ 0.01 - 0.001
- $p - value \leq 0.05$ and $> 0.01$
- **Bolded** when $r_s \geq 0.4$

[a] Furandiones correspond to the sum of 2,5-furandione (FD), dihydro-2,5-furandione(DFD), 3-methyl-2,5-furandione(MFD), and dihydro-3-methyl-2,5-furandione(DMFD).

[b] Hopanes correspond to the sum of 17α(H)-22,29,30-trisnorhopane, 17β(H)-21α(H)-30-norhopane, and 17α(H)-21β(H)-hopane.

Table S4: Daily measurements of organic carbon (OC), elemental carbon (EC), ambient temperature and relative humidity (RH), in Iowa City, IA from Nov 4 - Nov. 17, 2011.

| | OC (µg m-3) Average ± SD | EC (µg m-3) Average ± SD | Average daily Temperature (°C) | RH (%) |
|---|---|---|---|---|
| 11/4/2015 | 1.76 ± 0.10 | 0.14 ± 0.02 | 14 | 87 |
| 11/5/2015 | 0.66 ± 0.05 | 0.11 ± 0.02 | 15 | 85 |
| 11/6/2015 | 1.12 ± 0.07 | 0.11 ± 0.02 | 7 | 72 |
| 11/7/2015 | 2.28 ± 0.12 | 0.25 ± 0.02 | 5 | 68 |
| 11/8/2015 | 1.95 ± 0.11 | 0.26 ± 0.02 | 5 | 68 |
| 11/9/2015 | 2.90 ± 0.15 | 0.38 ± 0.03 | 5 | 65 |
| 11/10/2015 | 3.01 ± 0.16 | 0.26 ± 0.02 | 6 | 70 |
| 11/11/2015 | 1.95 ± 0.11 | 0.13 ± 0.02 | 11 | 81 |
| 11/12/2015 | 1.07 ± 0.07 | 0.03 ± 0.02 | 6 | 74 |
| 11/13/2015 | 1.56 ± 0.09 | 0.18 ± 0.02 | 4 | 57 |
| 11/14/2015 | 2.12 ± 0.12 | 0.13 ± 0.02 | 8 | 66 |
| 11/15/2015 | 9.65 ± 0.49 | 0.81 ± 0.04 | 10 | 59 |
| 11/16/2015 | 1.37 ± 0.08 | 0.10 ± 0.02 | 9 | 72 |
| 11/17/2015 | 1.23 ± 0.08 | 0.05 ± 0.02 | 13 | 95 |

Figure S1: Mass spectrum of phthalic anhydride (a) and phthalic acid (b), injected directly to a GCMS system equipped with a Carbowax column and an inlet heated to 250 °C. Both species eluted at the same retention time (13.46 min) and exhibited the same mass spectra, with $m/z$ 148 being the molecular ion for phthalic anhydride. Phthalic acid likely dehydrates (loses $H_2O$) in the GC the injection port forming its anhydride derivative. The trimethylsilylated derivatives mass spectra for phthalic anhydride (c) and phthalic acid (d) also had identical retention times (13.45 min) and exhibited identical mass spectra, with $m/z$ 310 being the molecular ion for the derivatized product.

Figure S2: The daily distribution of the detected nitromonoaromatics between gas and particle phases. The phase distributions of 4-nitrocatechol and 4-methyl-5-nitrocatechol are not shown in this figure due to the low extraction efficiency.

Figure S3: Daily gas and particle concentrations of select nitroaromatics detected in Iowa City during fall 2015, including 4-nitrophenol (4NP), 4-methyl-2-nitrophenol (4M-2NP), 2-methyl-4-nitrophenol (2M-4NP), 4-nitroguaiacol (4NG), and 4-hydroxy-3-nitrobenzyl alcohol (4H-3NB).

Figure S4: Ambient concentrations for individual furandiones detected in gas and particle phases. Furandiones were not detected in gas phase on 4 November. Gas phase concentrations were corrected to the average recovery (n=3) of authentic standards from the PUF (38.4 ± 2.8% for DFD, 31.8± 2.6% for MFD, and 44.2± 2.7% for DMFD).

**Works Cited**

EPA: Estimation Programs Interface Suite™ for Microsoft® Windows, v 4.11 or insert version used. United States Environmental Protection Agency, Washington, DC, USA., in, 2012.

Kluwer: Spectral atlas of polycyclic aromatic compounds, Kluwer Academic Publishers Dordrecht, The Netherlands 1988.

Linstrom, P. J., Mallard, W.G. (Eds.): "Entropy and heat capacity data" in NIST chemistry webbook, NIST standard reference database number 69, National Institute of Standards and Technology, Gaithersburg MD, 20899, http://webbook.nist.gov. , 2005.

Seinfeld, J. H., and Pankow, J. F.: Organic atmospheric particulate material, Annu. Rev. Phys. Chem., 54, 121-140, doi:10.1146/annurev.physchem.54.011002.103756, 2003.

Yaws, C. L., and Satyro, Marco A.: Enthalpy of vaporization—Organic compounds, in: Thermophysical Properties of Chemicals and Hydrocarbons Lamar University, Beaumont, Texas. , 344, 2009.